# Extent of Mucosal Inflammation in Ulcerative Colitis Influences the Clinical Remission Induced by Vedolizumab

**DOI:** 10.3390/jcm9020385

**Published:** 2020-02-01

**Authors:** Patrizio Scarozza, Irene Marafini, Federica Laudisi, Edoardo Troncone, Heike Schmitt, Marco Vincenzo Lenti, Stefania Costa, Irene Rocchetti, Elena De Cristofaro, Silvia Salvatori, Ludovica Frezzati, Antonio Di Sabatino, Raja Atreya, Markus F. Neurath, Emma Calabrese, Giovanni Monteleone

**Affiliations:** 1Department of Systems Medicine, University of Rome “Tor Vergata”, 00133 Rome, Italy; scarozzapatrizio@gmail.com (P.S.); irene.marafini@gmail.com (I.M.); federica.laudisi@gmail.com (F.L.); troncone.edoardo@gmail.com (E.T.); elena_decr@hotmail.it (E.D.C.); silviasalvatori23@gmail.com (S.S.); ludovica.frezzati@me.com (L.F.); emma.calabrese@uniroma2.it (E.C.); 2Department of Medicine 1, Friedrich-Alexander-University of Erlangen-Nuremberg, 91054 Erlangen, Germany; Heike.Schmitt@uk-erlangen.de (H.S.); raja.atreya@uk-erlangen.de (R.A.); Markus.Neurath@uk-erlangen.de (M.F.N.); 3First Department of Internal Medicine, San Matteo Hospital Foundation, University of Pavia, 27100 Pavia, Italy; marco.lenti@unipv.it (M.V.L.); stefania.costa01@universitadipavia.it (S.C.); a.disabatino@smatteo.pv.it (A.D.S.); 4Statistical Office, Superior Council of Judiciary, 00185 Rome, Italy; irene.rocchetti@gmail.com

**Keywords:** inflammatory bowel disease, Crohn’s disease, alpha4beta7 integrin, biologics

## Abstract

Randomized controlled clinical trials and real-life observations indicate that less than 50% of patients with Crohn’s disease (CD) or ulcerative colitis (UC) respond to vedolizumab, a humanized monoclonal antibody that blocks the α4β7 integrin. Since α4β7-expressing lymphocytes mainly infiltrate the left colon, we assessed whether localization of CD and UC influences vedolizumab-induced remission. One hundred and eighty-one patients (74 CD and 107 UC) receiving vedolizumab in 3 referral centers were retrospectively evaluated for clinical remission at week 14. Demographic and clinical characteristics were compared between remitters and non-responders, and multivariable multinomial analysis was performed to identify predictors of remission. Remission was achieved in 17 CD (23%) and 34 UC (32%) patients, respectively. In CD, localization of the lesions did not influence clinical remission. In UC, the remitters had more frequently a distal/left-sided colitis (21/34, 62%) as compared to the non-responders (9/47, 19%), and extensive colitis was more frequent in the non-responders (38/47, 81%) than in the remitters (13/34, 38%). The multivariable multinomial analysis showed that distal/left-sided colitis was associated with a higher probability of clinical remission while extensive colitis was inversely associated with induction of remission. Data indicate that UC patients with distal or left-sided colitis are more likely to achieve remission than patients with extensive colitis following vedolizumab treatment.

## 1. Introduction

In the last decades, a better understanding of the molecular mechanisms, which drive the pathological process in the gut of patients with Crohn’s disease (CD) and ulcerative colitis (UC), the major forms of inflammatory bowel diseases (IBD), has facilitated the development of drugs promoting resolution of mucosal inflammation and healing of the gut lesions [1,2,3,4,5]. 

One such a drug is vedolizumab, a gut-selective humanized monoclonal antibody that binds to the α4β7 integrin and selectively reduces intestinal immune cell trafficking, thereby providing a safe and effective treatment option for patients with IBD [6,7,8,9,10,11,12]. Vedolizumab is recommended for patients with active CD and active UC who have not responded to corticosteroids, immunosuppressants, or tumor necrosis factor (TNF) blockers. Almost half of the IBD patients respond to vedolizumab, but more than one-third of them relapse within 12–36 months from the induction phase [13,14,15,16,17,18,19,20,21,22,23]. These findings have boosted intensive research aimed at identifying predictors of response to the drug [14,15,16,17,18,19,20,21,22,23,24,25,26,27,28,29,30,31]. It is now evident that prior exposure to anti-TNF α drugs associates with a reduced induction of clinical and endoscopic remission [14,15,16,17,18,19,20,21,22,23,24,25,26,27,28,29,30,31]. Both CD and UC patients with a severe clinical activity at baseline are less likely to respond to vedolizumab, as compared to patients with mild-to-moderate disease [14,15,16,17,18,19,20,21,22,23,24,25,26,27,28,29,30,31]. In addition to prior anti-TNF α therapy, active smoking, penetrant behavior, and active perianal disease at baseline are independent predictors of nonresponse to the drug in CD patients [19]. 

More recent studies have shown that, at least in UC, patients achieving and maintaining clinical and endoscopic remission have significantly higher vedolizumab trough concentration during maintenance therapy than patients who do not respond [32,33,34,35,36,37,38,39]. Nevertheless, it remains plausible that further clinical and biological factors can influence the response to the drug. Indeed, it has been reported that accumulation of α4β7-expressing immune cells in the gut tissue can predict therapeutic success of vedolizumab [40]. Notably, in the healthy intestine, α4β7-expressing T cells accumulate preferentially in the left colon, raising the possibility that localization of the active lesions can further influence response to vedolizumab [41,42,43,44]. 

This study was aimed at examining whether disease location predicts clinical remission to vedolizumab. 

## 2. Materials and Methods

### 2.1. Study Design and Aim

This was a retrospective study conducted on IBD patients treated with vedolizumab in three referral centers: Tor Vergata University Hospital (Rome, Italy), San Matteo Hospital of Pavia (Pavia, Italy), and Friedrich-Alexander-University Hospital of Erlangen (Erlangen, Germany). Patients’ data were retrospectively collected between April 2018 and October 2019 and, after a de-identification process, registered into an electronic database. The primary objective of the study was to assess whether the location of CD and UC influenced vedolizumab-induced remission at week 14. Moreover, we evaluated whether further clinical and demographic factors influenced both clinical response and remission at week 14. The study was approved by the local Ethics Committee (CEI Policlinico Tor Vergata, Rome).

### 2.2. Patients

Inclusion criteria included: a confirmed diagnosis of CD or UC [45,46]; a clinically active disease at baseline (regardless of the grade) requiring vedolizumab treatment; available data on clinical outcome at baseline and at week 14. Patients were excluded if they were in clinical remission at baseline, had unclassified/indeterminate colitis, or pouchitis and if the clinical data at the indicated time points were not available.

For each patient, several demographic and clinical variables were considered for the analysis (Table 1). Clinical disease activity for UC was evaluated by the partial Mayo (pMayo) score [47] and for CD by the Harvey Bradshaw index (HBI). [48] Clinical remission was defined as a pMayo score <2 (UC) and an HBI <5 (CD), while clinical response was defined as a reduction of minimum 3 points of pMayo score and HBI for UC and CD, respectively. Clinical evaluation of treatment response in the 3 participating centers took place at the same time using the same clinical scores above stated.

### 2.3. Statistical Analysis

Continuous variables were reported as median with interquartile range (IQR) and categorical variables were expressed as percentage. Distribution of the variables at baseline between the groups of comparison (remitters vs. non-responders and responders vs. non-responders) was evaluated with binomial analysis, using the χ^2^ or Fisher exact test. A multinomial logistic model for a constructed variable Y has been applied to assess the predictive factors of the clinical remission (Y1) and the clinical response (Y2) separately. The group of the non-responders was considered as the reference group for the binomial and multinomial logistic analysis. A *p* < 0.05 level was considered for statistical significance.

## 3. Results

### 3.1. Induction of Clinical Remission

One hundred and eighty-one IBD patients (74 CD and 107 UC) were enrolled. Twenty-two patients were excluded because their clinical data were not available. Patients had a median duration of disease longer than 10 years and most of them (85% of CD patients and 80% of UC patients) had been previously exposed to TNF α antagonists (Table 1). Most of the patients enrolled had a mild-to-moderate activity at baseline (Table 1). In CD, there was no statistical association between the clinical activity at baseline and disease location (Appendix A). Similarly, no association was seen between the clinical activity and behavior except for the stricturing phenotype, which was significantly associated with a moderate activity (Appendix A). In UC, the extent of the lesions was not associated with the clinical activity at baseline (Appendix A).

At week 14, 17/74 (23%) CD patients and 34/107 (32%) UC patients were in clinical remission (Figure 1A). In CD, a mild clinical activity at baseline was significantly more frequent in the group of remitters (11/17, 65%) than in the group of the non-responders (7/40, 18%; *p* = 0.0004) (Appendix A), while a moderate clinical activity was less frequent in patients with clinical remission (6/17, 35%) than in the non-responders (31/40, 77%; *p* = 0.002). There was no difference between remitters and non-responders for the remaining demographic and clinical variables, as well as for the prior or current use of drugs (Appendix A).

In UC, severe clinical activity at baseline was documented in 9/ 47 (19%) non-responders and in no patient achieving clinical remission (*p* = 0.008) (Appendix A). Moreover, the remitters had greater frequency of distal/left-sided colitis (21/34, 62%) as compared to the non-responders (9/47, 19%; *p* = 0.00008). In contrast, extensive colitis was more frequent in the non-responders (38/47, 81%) than in the remitters (13/34, 38%; *p* = 0.00008) (Appendix A). No further association was observed between each of the two groups of remitters and non-responders and the remaining demographic and clinical variables or the prior or current use of drugs (Appendix A).

### 3.2. Predictive Factors of Remission

The multinomial analysis revealed that, in CD, induction of clinical remission was significantly associated with male gender and a stricturing disease behavior, while there was an inverse association between the clinical activity of the disease at baseline and induction of remission (Table 2). No association between the other demographic and clinical variables and induction of remission was found (Table 2). Similarly, the prior use of TNF α antagonists, the prior or current use of immunosuppressors, and the concomitant use of steroids, as well as the baseline level of serum CRP did not associate with induction of remission (Table 2).

In the UC group, distal/left-sided colitis was significantly associated with induction of remission (*p* = 0.0003) (Table 2). In contrast, extensive colitis and clinical activity at baseline were inversely associated with remission (*p* = 0.0003 and *p* = 0.037, respectively) (Table 2). There was no further association between induction of remission and the remaining variables considered in the study (Table 2).

### 3.3. Clinical Response and Predictors

Clinical response was achieved in 17/74 (23%) CD patients and in 26/107 (24%) UC patients (Figure 1B). For both CD and UC, all the considered variables at baseline were equally distributed between responders and non-responders (Appendix A).

In CD, the multivariable multinomial analysis showed an association between the concomitant use of immunosuppressors during the induction phase with vedolizumab and clinical response while an inverse association was seen between the penetrating behavior of the disease and clinical response (Table 3). No association between clinical response and the remaining variables considered in this study was seen (Table 3). In UC, there was no association between clinical response and any of the variables analyzed (Table 3). Mild activity of the disease at baseline was less frequent in the group of responders (4/26; 15%) than in the group of remitters (13/34; 38%, *p* = 0.05), while moderate-to-severe disease was more common in the group of responders (22/26; 85%) than in the group of remitters 21/34; 62%, *p* = 0.05).

## 4. Discussion

This study aimed to identify clinical predictors of response to vedolizumab. Through a retrospective analysis of clinical data of patients receiving the drug in 3 referral centers, we initially showed that induction of remission at week 14 was achieved in more than one fifth of CD patients and nearly one third of UC. Additionally, clinical response at the same time point was seen in another one fifth of CD patients and UC patients, thus confirming previous real-life studies showing that nearly half of the IBD patients can benefit from vedolizumab treatment [14,15,16,17,18,19,20,21,22,23,24,25,26,27,28,29,30,31]. Altogether, these findings indicate the need to identify predictive factors of response to the drug in order to optimize the therapeutic strategy.

In addition to the classical demographic and clinical variables considered in previous studies, we herein included the impact of intestinal location of disease on the induction of remission, as response to the treatment is strictly dependent on the number of α4β7-positive cells in inflamed gut and it is known that such cells are more frequent in the distal parts of the colon [43]. Our findings indicate that, in CD, induction of remission occurred more frequently in male, in line with data emerging from the GEMINI 2 study [11]. Surprisingly, stricturing behavior was associated with a greater probability of achieving clinical remission. The reasons why patients with strictures should benefit from vedolizumab treatment more than patients with other phenotypes remain unknown. A possibility is that such patients had a mild inflammatory process overlying the strictured area, which could be dampened by vedolizumab treatment. This hypothesis would be consistent with the demonstration that patients with a mild active disease respond better than patients with a more severe disease [14,15,16,17,18,19,20,21,22,23,24,25,26,27,28,29]. This later finding emerges also from our analysis showing an inverse association between induction of remission and clinical activity of the disease at baseline. Another possibility is that vedolizumab can interfere with signaling pathways involved in the fibrogenetic process. In this context, it is noteworthy that α4β7 integrin, the molecular target of vedolizumab, could also bind to fibronectin, a component of the extracellular matrix abundantly produced in intestinal strictures and supposed to enhance collagen secretion [49,50]. Further work is warranted to address these issues.

No association was found between induction of remission and the remaining variables analyzed in the study. In particular, our data did not confirm previous studies reporting an inverse association between the prior use of TNF α blockers and response to vedolizumab [19,20]. The reason for such a discrepancy remains unclear, but it is conceivable that our data could be somehow biased by the fact that only a minority of the patients (15%) was naïve to TNF α antagonists. Similarly, smoking, active perianal disease, location of the disease, and abnormal CRP at baseline, which appeared to influence response to the treatment in some studies, were not associated with induction of remission in our study [15,19,20,51].

Our data indicate that, in the CD cohort, there was a positive association between concomitant immunosuppressive therapy during the induction phase and a better clinical outcome at week 14, probably reflecting the ability of vedolizumab and thiopurines to suppress distinct inflammatory pathways, which sustain the pathologic process in CD.

Notably, a different scenario emerged when UC-related data were analyzed, as extent of mucosal lesions largely influenced induction of remission. Indeed, the remitters had more frequently a distal/left-sided colitis as compared to the non-responders. The multivariable multinomial analysis showed that distal/left-sided colitis was associated with a higher probability of clinical remission while extensive colitis was inversely associated with induction of remission. This association was independent of the baseline disease activity. These data are in line with those of a recent study showing that distal/left-sided colitis was associated with a higher percentage of clinical remission at week 14 as compared to extensive colitis following vedolizumab treatment [14]. However, other studies failed to document an association between extent of UC and response to vedolizumab [22,23,52]. Although, it remains unclear why such studies have generated divergent results, it is conceivable that discrepancies can rely on differences in the selection of the objectives, study population, and statistical analysis adopted.

Despite patients with distal/left-sided colitis were more likely to achieve remission following vedolizumab treatment, no association was seen between the induction of clinical response and extent of colitis. This could in part rely on the different grade of activity of the disease at baseline in the groups of remitters and responders. Indeed, remitters had a higher frequency of mild disease as compared to responders while moderate-to-severe disease was more frequent in the group of responders.

This study has important strengths, such as the multicentric origin of the data and the novelty of the primary objective. However, the retrospective nature of the data and the lack of information on endoscopic/histological response to the treatment represent major limits. As mentioned above, the majority of the patients was already exposed to anti-TNF therapy, as vedolizumab at the time-point of investigation was mainly used in patients who were unresponsive or intolerant to TNF blockers in the examined centers. Therefore, we are aware that the present data deserve further confirmation prior to be generalized to the whole IBD population.

## 5. Conclusions

In conclusion, we here show that the extent of mucosal inflammation in UC is a major determinant of clinical remission induced by vedolizumab. However, prospective studies on large populations investigating clinical and endoscopic/histological remission are needed to confirm such data.

## Figures and Tables

**Figure 1 jcm-09-00385-f001:**
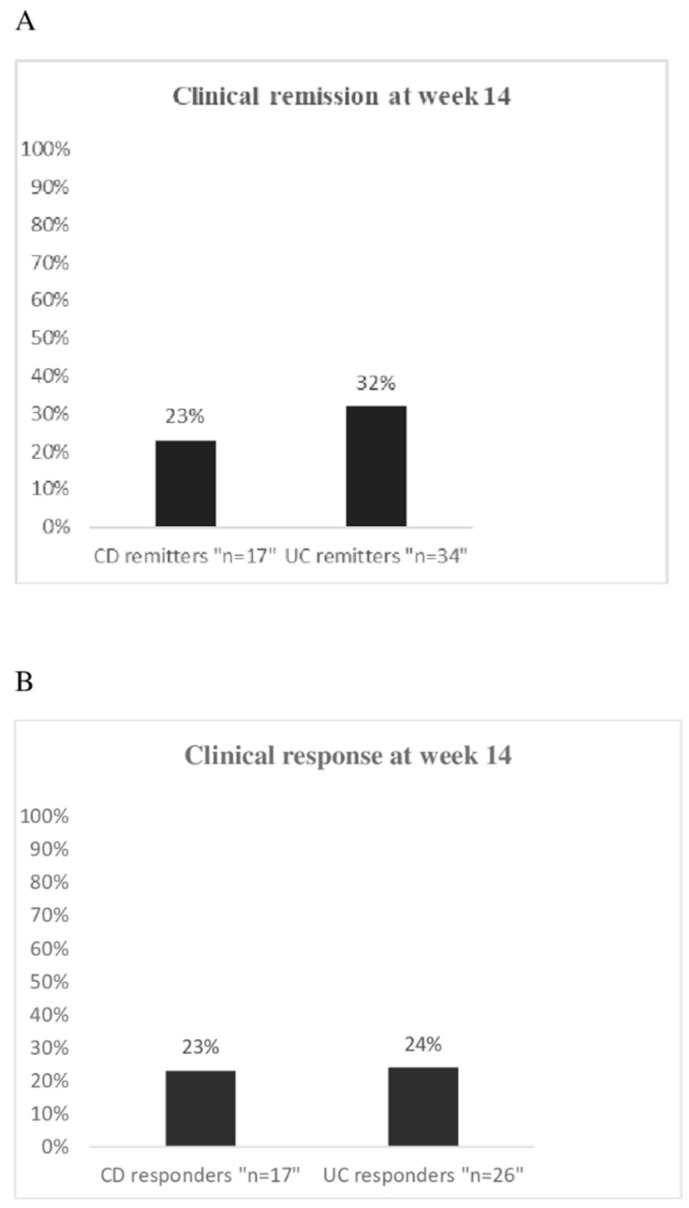
(**A**) Percentage of clinical remission in 74 CD patients and 107 UC patients evaluated at week 14 upon vedolizumab treatment; (**B**) Percentage of clinical response in 74 CD patients and 107 UC patients evaluated at week 14 upon vedolizumab treatment.

**Table 1 jcm-09-00385-t001:** Clinical and demographic characteristics of the patients.

Variable	Crohn’s Disease (*n* = 74)	Ulcerative Colitis (*n* = 107)
median age, years (IQR)	42 (33–55)	46 (32–56)
median disease duration from IBD diagnosis, years (IQR)	14 (10–23)	11 (7–19)
male gender, *n* (%)	34 (46%)	56 (52%)
smoking status, *n* (%)		
never	34 (46%)	74 (69%)
former	19 (26%)	19 (18%)
current	21 (28%)	14 (13%)
Montreal disease location, *n* (%)		
L1 (ileal disease)	20 (27%)	
L2 (colonic disease)	7 (9%)	
L3 (ileo-colonic disease)	47 (64%)	
E1 (proctitis)		3 (3%)
E2 (left-sided colitis)		37 (34%)
E3 (extensive colitis)		67 (63%)
upper disease location, *n* (%)	15 (20%)	
Montreal disease behavior, *n* (%)		
B1 (non-stricturing, non-penetrating)	24 (32%)	
B2 (stricturing)	23 (31%)	
B3 (penetrating)	27 (37%)	
Mild Clinical Activity	26 (35%)	31 (29%)
Moderate Clinical Activity	45 (61%)	64 (60%)
Severe Clinical Activity	3 (4%)	12 (11%)
perianal disease, *n* (%)	23 (31%)	
prior ileo-colonic resection, *n* (%)	44 (59%)	
prior TNF antagonists, *n* (%) *	63 (85%)	86 (80%)

IQR: Interquartile range. Mild Clinical Activity (HBI 5–7 for CD patients and pMayo 2–4 for UC patients). Moderate Clinical Activity (HBI 8–16 for CD patients and pMayo 5–7 for UC patients). Severe Clinical Activity (HBI >16 for CD patients and pMayo >7 for UC patients). * TNF antagonists were discontinued for primary non-response or intolerance to the drug.

**Table 2 jcm-09-00385-t002:** Predictive factors of clinical remission at week 14 in Crohn’s disease and ulcerative colitis patients.

Variable	Estimate (CD)	*p*-Value (CD)	Estimate (UC)	*p*-Value (UC)
prior anti-TNF	6.293	0.111	2.298	0.259
prior immunosuppressive therapy	1.563	0.734	0.927	0.910
concomitant steroids	3.126	0.265	0.889	0.838
concomitant immunosuppressive therapy	17.467	0.112	2.296	0.437
male gender	59.636	0.010	1.983	0.227
CRP > 5 mg/L	5.428	0.122	1.396	0.558
current smoker	3.206	0.361	1.210	0.831
ex smoker	0.283	0.357	0.976	0.976
clinical activity	0.356	0.0007	0.676	0.037
disease duration from IBD diagnosis	1.144	0.062	0.987	0.663
colonic disease	2.941	0.445		
Ileo-colonic disease	8.801	0.148		
Distal/left-sided colitis			2.154	0.0003
Extensive colitis			0.116	0.0003
upper disease	4.776	0.297		
stricturing disease	22.079	0.047		
penetrating disease	1.709	0.727		
perianal disease	1.507	0.743		
prior ileo-colonic resection	0.715	0.802		

Continuous variables: clinical activity; disease duration. All others are categorical variables.

**Table 3 jcm-09-00385-t003:** Predictive factors of clinical response at week 14 in Crohn’s disease and ulcerative colitis patients.

Variable	Estimate (CD)	*p*-Value (CD)	Estimate (UC)	*p*-Value (UC)
prior anti-TNF	0.118	0.411	2.111	0.341
prior immunosuppressive therapy	8.920	0.159	1.144	0.830
concomitant steroids	3.705	0.187	0.814	0.721
Concomitant immunosuppressive therapy	21.030	0.024	4.533	0.121
male gender	1.942	0.484	0.714	0.537
CRP >5 mg/L	4.276	0.153	0.742	0.595
current smoker	1.268	0.813	2.494	0.250
ex smoker	0.398	0.491	1.349	0.692
clinical activity	0.687	0.021	1.052	0.798
disease duration from IBD diagnosis	0.996	0.953	0.951	0.156
colonic disease	3.610	0.445		
Ileo-colonic disease	0.768	0.814		
Distal/left-sided colitis			1.081	0.084
Extensive colitis			0.339	0.084
upper disease	2.712	0.403		
stricturing disease	1.352	0.791		
penetrating disease	0.033	0.037		
perianal disease	7.565	0.133		
prior ileo-colonic resection	0.756	0.828		

Continuous variables: clinical activity; disease duration. All others are categorical variables.

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
