# Peer review of "Extent of Mucosal Inflammation in Ulcerative Colitis Influences the Clinical Remission Induced by Vedolizumab"

_jcm, 2020, doi:10.3390/jcm9020385_

Round 1

Reviewer 1 Report

This study addresses the novel hypothesis that anatomic distribution of disease, specifically location in distal or left-sided colon, might be a major determining factor in the degree of responsiveness to vedolizumab. A rationale is offered in terms of the distal distribution of alpha-4 beta-7 receptors in the colon.

Support for this hypothesis clearly depends upon the control for potential confounding factors. The authors have carefully sought this control by their multinomial logistic analysis. Despite this meticulous effort, however, I am not convinced that they have distinctly separated the twin effects of disease extent and clinical severity. (See lines 109-113, 131-132, 166-168.)

There is one other potentially confounding factor that the authors do not appear to have taken into consideration. UC cases with proximal extension from previously left-sided disease may constitute a particular phenotype. (See Roda G et al, AP&T 2017; Sahami S et al, United Eur Gatroenterol J 2017; Burisch J et al, J Crohn's Colitis 2017; Qiu Y et al, BMC Gastroenterol 2019.) Since up to one-third of left-sided UC patients may fall into this category, it might be useful to know whether the results were any different in this particular phenotype.

Author Response

We thank the reviewer for his/her helpful comments and suggestions. In response to the specific points raised by the reviewer:

This study addresses the novel hypothesis that anatomic distribution of disease, specifically location in distal or left-sided colon, might be a major determining factor in the degree of responsiveness to vedolizumab. A rationale is offered in terms of the distal distribution of alpha-4 beta-7 receptors in the colon

We thank the reviewer for the positive assessment of our scientific findings.

Support for this hypothesis clearly depends upon the control for potential confounding factors. The authors have carefully sought this control by their multinomial logistic analysis. Despite this meticulous effort, however, I am not convinced that they have distinctly separated the twin effects of disease extent and clinical severity. (See lines 109-113, 131-132, 166-168.)

This is a very good point. We took on board such a suggestion and analyzed distinctly the effect of disease extent and clinical severity on the response to the drug. To this end, we compared the distribution of clinical activity (mild, moderate and severe) of the disease between the group of patients with distal/left-sided colitis and the group of patients with extensive colitis. No statistical difference was found between the two groups of patients (see supplementary table S3).

There is one other potentially confounding factor that the authors do not appear to have taken into consideration. UC cases with proximal extension from previously left-sided disease may constitute a particular phenotype. (See Roda G et al, AP&T 2017; Sahami S et al, United Eur Gatroenterol J 2017; Burisch J et al, J Crohn's Colitis 2017; Qiu Y et al, BMC Gastroenterol 2019.) Since up to one-third of left-sided UC patients may fall into this category, it might be useful to know whether the results were any different in this particular phenotype.

We agree with reviewer that left-sided UC can extend proximally over the time but in honesty we do not believe this could impact on the main findings of the study, as we tested the hypothesis that extent of the lesions at the time of Vedolizumab treatment can influence the effectiveness of the drug.

Reviewer 2 Report

The authors demonstrated UC patients with distal/left-sided colitis are more likely to achieve remission than patients with extensive colitis by Vedolizumab treatment. This is very interesting result, however I can not find the original article which showed α4β7-expressing T cells accumulate preferentially in the left colon. I raise several concerns including this point as listed below.

The authors mentioned that ‘Notably, in the healthy intestine, α4β7-expressing T cells accumulate preferentially in the left colon, raising the possibility that localization of the active lesions can further influence response to Vedolizumab.’ I can not find the original article which showed α4β7-expressing T cells accumulate preferentially in the left colon. Please refer the original article and describe it in detail.

The authors showed that clinical remission ratio by Vedolizumab treatment was higher than clinical response ratio. Did this result mean that there were many mild cases in this study? The authors need to show the characteristic of disease severity in Table1.

Furthermore, there was a significantly high clinical remission ratio in the distal/left side colitis, however there was no significant difference in the locations in clinical response rate. How do the authors explain the discrepancy of these results? The authors also need to show the data, which clary the association between disease location/behavior and disease severity in Table.

In CD, the multivariable multinomial analysis showed a significant association between the concomitant use of immunosuppressors by Vedolizumab treatment and clinical response. The authors need to discuss the usefulness of the imunosupressors with Vedolizumab treatment of CDs in Discussion.

I can not find Figure 1.

Author Response

Reviewer #2:

We thank the reviewer for his/her helpful comments and suggestions. In response to the specific points raised by the reviewer:

The authors demonstrated UC patients with distal/left-sided colitis are more likely to achieve remission than patients with extensive colitis by Vedolizumab treatment. This is very interesting result, however I can not find the original article which showed α4β7-expressing T cells accumulate preferentially in the left colon. I raise several concerns including this point as listed below.

Reference 43 refers to the original article showing distribution of a4b7-expressing T cells in the different colonic tracts

Smids C, Horjus Talabur Horje CS, van Wijk F, et al. The Complexity of alpha E beta 7 Blockade in Inflammatory Bowel Diseases. J Crohns Colitis. 2017;11:500-508.

The authors showed that clinical remission ratio by Vedolizumab treatment was higher than clinical response ratio. Did this result mean that there were many mild cases in this study? The authors need to show the characteristic of disease severity in Table1

This is valid point. The fact that the remission rate by Vedolizumab treatment was higher than the response rate could be somehow related to the higher frequency of mild-to-moderate activity in the group of remitters as compared to the groups of responders.

As suggested by the reviewer, the characteristics of disease severity are shown in Table 1.

Furthermore, there was a significantly high clinical remission ratio in the distal/left side colitis, however there was no significant difference in the locations in clinical response rate. How do the authors explain the discrepancy of these results? The authors also need to show the data, which clary the association between disease location/behavior and disease severity in Table.

As correctly pointed-out by the reviewer localization of the disease influenced the rate of clinical remission but not the rate of clinical response. As pointed out above, this could rely on the fact that mild activity of the disease at baseline was less frequent in the group of responders (4/26; 15%) than in the group of remitters (13/34; 38%, p=0.05), while moderate-to-severe disease was more common in the group of responders (22/26; 85%) than in the group of remitters 21/34; 62%, p=0.05). This has been made clear in the results and discussion sections.    

We took on board the reviewer’s suggestions and provided data about the association between disease location/behaviour and disease severity (Table S1, S2, S3).

In CD, the multivariable multinomial analysis showed a significant association between the concomitant use of immunosuppressors by Vedolizumab treatment and clinical response. The authors need to discuss the usefulness of the imunosupressors with Vedolizumab treatment of CDs in Discussion.

As suggested by the reviewer, we have revised the discussion section and discussed the usefulness of the imunosupressors with Vedolizumab in CD.

I can not find Figure 1.

We are sorry for this. However, Figure 1 was up-loaded with the text during the submission process of the original manuscript.

Reviewer 3 Report

Brief summary: This paper with data from three European referral centers between April 2018 and October 2019, contributes with interesting findings regarding the influence of disease extension on remission of IBD patients in treatment with Vedolizumab.

Broad Comments: The paper is concise and the title appropriate. The introduction provides a sufficient background of the topic and the aim is well defined. The results are provided in a structured manner and the discussion includes important results of previous research within the field. However, clinical activity as well as remission are based on scores rather than biochemical and fecal markers or endoscopy which is a disadvantage when evaluating the results. Moreover, as a vast majority of the patients (80%) were previously exposed to anti-TNF treatment no firm conclusions can be drawn from the present study.

Introduction:

no comments

Materials and Methods:

A list of reasons for discounting anti-TNF A flowchart of the study population including reasons and number of excluded patients would be appreciated in order to evaluate the accuracy of the study population. Since the results are based on retrospectively retrieved clinical data from three different centers in two different countries, there is a possibility that the treatment response is evaluated differently. If there is a consensus that the clinical evaluation of treatment response took place at the same time and that the regimes was completely similar in all three centers it would facilitate if it appeared more clearly in the text. A statement regarding ethics is lacking

Results:

Clarification/definition is needed for: Altered CRP at baseline, disease duration as well as prior surgery A suggestion would be to reduces the decimals to just one or two.

Discussion:

Would it be possible to distribute the text into smaller segments to increase the understanding and maybe add subheadings such as: Main findings, clinical implications, comparisons with previous research.

Specific comments:

P 3, line 113-114: the sentence dose not completely make sense. “The two groups of remitters and non-responders”? if you mean that there was no difference between remitters and non-responders, please rephrase.

P 3, line119-121: the sentence dose not completely make sense. “The two groups of remitters and non-responders”? if you mean that there was no difference between remitters and non-responders, please rephrase.

P 4, line 135. In table 2, since all values are at baseline, the variable “altered CRP at baseline” could be shortened to Altered CRP

P 5, line 153; one third

P 5, line 153-154; Add more than one fifth of CD patients…

Supplementary file:

Table s1 it would facilitate if there were 3 columns instead of 2; CD remitters, CD responders, CD non-responders. Define abnormal CRP

Author Response

We thank the reviewer for his/her helpful comments and suggestions. In response to the specific points raised by the reviewer:

Broad Comments: The paper is concise and the title appropriate. The introduction provides a sufficient background of the topic and the aim is well defined. The results are provided in a structured manner and the discussion includes important results of previous research within the field. However, clinical activity as well as remission are based on scores rather than biochemical and fecal markers or endoscopy which is a disadvantage when evaluating the results. Moreover, as a vast majority of the patients (80%) were previously exposed to anti-TNF treatment no firm conclusions can be drawn from the present study.

We had already taken into consideration and widely discussed the issues raised by this reviewer. In this context, we would like to point out that prospective studies are now ongoing in our centers to confirm the major findings of this study.  

Materials and Methods:

A list of reasons for discounting anti-TNF A flowchart of the study population including reasons and number of excluded patients would be appreciated in order to evaluate the accuracy of the study population.

We have specified in the Table 1 that TNF antagonists were discontinued for primary non-response or intolerance to the drug. Moreover, in the methods section, it was already made clear that patients were excluded if they were in clinical remission at baseline, had unclassified/indeterminate colitis, or pouchitis and if the clinical data at the indicated time points were not available. In the results section, we have added a sentence to indicate the number of patients excluded and the reason why those patients were excluded. Since, the number of such patients was low, no flowchart of the study population was included.

Since the results are based on retrospectively retrieved clinical data from three different centers in two different countries, there is a possibility that the treatment response is evaluated differently. If there is a consensus that the clinical evaluation of treatment response took place at the same time and that the regimes was completely similar in all three centers it would facilitate if it appeared more clearly in the text.

In the methods section, it was made clear that clinical evaluation of treatment response in the 3 participating centers took place at the same time using the same clinical scores.

A statement regarding ethics is lacking.

We have specified in the text that the study was approved by the local Ethics Committee.

Results:

Clarification/definition is needed for: Altered CRP at baseline, disease duration as well as prior surgery A suggestion would be to reduces the decimals to just one or two.

We made the suggested corrections.

Discussion:

Would it be possible to distribute the text into smaller segments to increase the understanding and maybe add subheadings such as: Main findings, clinical implications, comparisons with previous research.

We have revised the discussion section and divided the text in smaller segments.

Specific comments:

 P 3, line 113-114: the sentence dose not completely make sense. “The two groups of remitters and non-responders”? if you mean that there was no difference between remitters and non-responders, please rephrase.

We have made the requested changes.

P 4, line 135. In table 2, since all values are at baseline, the variable “altered CRP at baseline” could be shortened to Altered CRP

We have made the requested changes.

P 5, line 153; one third

We have made the requested changes.

P 5, line 153-154; Add more than one fifth of CD patients…

We have made the requested changes.

Supplementary file:

Table s1 it would facilitate if there were 3 columns instead of 2; CD remitters, CD responders, CD non-responders. Define abnormal CRP

We took the decision to keep the original Tables1, as this is the only way to calculate differences between remitters and non-responders. Adding another column would make difficult ascertain any association between groups.

Round 2

Reviewer 1 Report

I appreciate the extensive efforts to which the authors have gone to address the question of disease severity as a possible confounding factor. I do believe they have not fully understood my second point, which is that “extensive” colitis comprises at least two different entities with different inherent prognoses: cases that were extensive in from the onset and cases that had spread proximally from the rectosigmoid only late in the course. Be that as it may, the efforts to correct for different activity levels at time of vedolizumab treatment have ameliorated some of these concerns.

Author Response

I appreciate the extensive efforts to which the authors have gone to address the question of disease severity as a possible confounding factor. I do believe they have not fully understood my second point, which is that “extensive” colitis comprises at least two different entities with different inherent prognoses: cases that were extensive in from the onset and cases that had spread proximally from the rectosigmoid only late in the course. Be that as it may, the efforts to correct for different activity levels at time of vedolizumab treatment have ameliorated some of these concerns.

 We understood the valid comments of the reviewer: indeed, we were aware that extensive” colitis comprises at least two different entities with different inherent prognoses. However, independently of that, we still believe that localization at the time of Vedolizumab treatment and not at the time of UC onset influences the response to the drug due to the different anatomical distribution of alpha4beta7 cells in the colon (e.g. predominant accumulation of the cells in the left colon). Anyway, we appreciate the reviewer’ s suggestion but, unfortunately, we have no sufficient clinical data to answer properly the reviewer's question. In fact, in the majority of the patients enrolled in the study, the initial diagnosis of UC was made in other hospitals and reports about the initial extent of the disease were missing or unclear (e.g. very often biopsy samples were not taken from the apparently unaffected areas, there was neither description about the histology of the samples nor information about the extent of the lesions).

Reviewer 2 Report

The authors have addressed most of my comments. However, one question remains. Reference 43 is a review article, but not original article. Please refer the original article, which showed the distribution of α4β7-expressing T cells in human intestine.

Author Response

We thank the reviewer for his/her suggestions. In response to the specific points raised by the reviewer:

The authors have addressed most of my comments. However, one question remains. Reference 43 is a review article, but not original article. Please refer the original article, which showed the distribution of α4β7-expressing T cells in human intestine.

We thank the reviewer for the suggestion. In fact, we had mistakenly reported a review article instead of the original paper showing a different expression of alpha4 beta7 integrin in the various parts of the gut. Therefore, we have changed the reference n.43. The new one is: Elewaut D, Van Damme N, De Keyser F, Baeten D, De Paepe P, Van Vlierberghe H, Mielants H, Cuvelier C, Verbruggen G, Veys EM, De Vos M. Altered expression of alpha E beta 7 integrin on intra-epithelial and lamina propria lymphocytes in patients with Crohn's disease. Acta Gastroenterol Belg. 1998 Jul-Sep;61(3):288-94.

Reviewer 3 Report

the manuscript has been significantly improved

Author Response

We thank the reviewer for his/her positive evaluation